# Quasi-Guided Modes Supported by a Composite Grating Structure with Alternating Element Widths

**Min Sun and Zhanghua Han \***

Shandong Provincial Key Laboratory of Optics and Photonic Devices, Center of Light Manipulation and Applications, School of Physics and Electronics, Shandong Normal University, Jinan 250358, China
* Correspondence: zhan@sdnu.edu.cn

**Abstract:** The efficiency of many optical processes is significantly dependent on the magnitude of the electric field. In this context, many artificially made resonating structures have been investigated to enhance light–matter interactions and facilitate the creation of practical applications. While metal–based terahertz metamaterials have been extensively investigated for this purpose, their performances are mainly limited by the poor confinement of terahertz waves on metal surfaces, exhibiting low resonance quality factors. In this work, we propose and investigate a simple yet novel scheme of enhancing wave–matter interactions in the terahertz region by exploiting the phenomenon of quasi–guided modes. The quasi-guided modes with ultra–high quality factors and huge local field enhancement can be achieved by manipulating the guided modes supported by a slab waveguide. The guided modes with the dispersion lines below the light line have infinite Q factors and can not be accessed from external space. By using a new type of composite grating composed of two ridge grating arrays with alternating ridge widths, the grating period is doubled, leading to a folding of the first Brillouin Zone and the flipping of the dispersion lines to be above the light line. Then, the guided modes will be transitioned into new quasi–guided modes with the possibility of free–space excitation while the Q factors are determined by the level of period–doubling perturbation. The presented results of realizing quasi–guided modes can be extended to other structures, providing a novel means of manipulating light–matter interactions.

**Keywords:** high quality (Q) factor; quasi-guided modes; guided mode resonances

## 1. Introduction

Many applications across the whole electromagnetic (EM) spectrum rely on the interaction between EM waves and matter, whose efficiency is highly dependent on the magnitude of electric field. Due to this reason, many processes are not efficient enough in many regions of the EM spectrum, especially when the sources are less developed and high−power optical sources are absent [1,2]. Typical examples are in the band of terahertz (THz) and Gigahertz (GHz), which has attracted broad research interest in recent years in both fundamental disciplines such as physics, chemistry and astronomy as well as in applied sciences including security checking, bio−medical diagnosis, gas detection, environmental monitoring and semiconductor industry, etc. [3–7], due to the unique properties of spectral−resolving capability, high transmission in many optically opaque dielectrics and low photon radiation energy. In particular, many chemicals have unique absorption spectral features arising from the molecular vibration or rotations, and these spectral fingerprints can be used for the detection and identification of unknown samples, including the typical examples of toxic gases [8] and drugs. Compared to electromagnetic waves in the visible and infrared bands, THz waves have much longer wavelengths, suggesting the requirement of correspondingly larger optical path to get a similar level of drop in the transmittance. On the other hand, most samples generally have a high−level of absorption resulting from the spectral overlap of multiple absorption resonances in the THz band.

This situation is more significant for biological samples where water is always present. As a result, a thin level of sample thickness is always preferred to be used. Unfortunately, the fact that most terahertz sources, especially those based on the optical generation method, have relatively weak output powers, further aggravates this situation from the point of view of signal−to−noise ratios. These conditions impose stringent requirements on the use of artificial structures to enhance THz–matter interactions.

In the optical frequencies, it has long been accepted that artificial structures can be used to enhance light–matter interactions, and then many exotic phenomena will be observed even when the optical input is weak [9–11]. Nanoscale optical elements in the form of surface plasmon−based metallic nanoantennas [10] or all−dielectric nanostructures [12] have been mainly explored, with demonstrated success in a large variety of applications such as enhanced spontaneous emission [13], nonlinear optics with alleviated phase matching condition [14], and photodetection [15], to name just a few. Unfortunately, in the THz band the electromagnetic characteristics of most metals are quite different from the optical frequencies, e.g., the conductivities are extremely high (~$10^7$ S/m), leading to a near−zero skin depth of the THz waves in metals and a loosely confined surface mode. That is why most metal−based metamaterials in the terahertz band or even lower frequencies have extremely low quality factors at the order of only 10, although the material loss is relatively low for metals in these frequencies. To this end, the concept of spoof surface plasmon (SSP) has been introduced into THz optics [16], and THz stripe antennas have been proposed to have a confined mode with large local field enhancement [17]. However, the requirement of periodic metal corrugations to support the SSP mode has limited the versatile applications in many circumstances.

One of the main attributes used in characterizing a resonating structure is its quality (Q) factor, which measures the damping rate of the resonance mode. While the Q factor can be straightforwardly quantified by the spectral width of the resonance, it is intrinsically related to other characteristics of the resonator, e.g., the maximum field enhancement [18]. As a result, to achieve an optimal THz–matter interaction, one aims at designing a resonating structure with finite yet high quality factors. For instance, Ranjan Singh et al. studied a Fano–type resonance in a two–gap split ring resonator composed of two asymmetric metal arcs, and they demonstrated a low Q factor of only 50 in the THz band [19]. In order to achieve metamaterials with high Q factors in the THz band, the main barrier is the radiative loss which should be reduced so as to increase the $Q_{rad}$. One effective method to achieve this goal in the optical frequencies is the exploration of the concept of bound states in the continuum (BICs), which was first proposed in quantum mechanics [20] and then intensively investigated in nanophotonics only in recent years to achieve ultra−high Q resonances [21–24]. Optical BICs by employing the grating structures on a slab have also been investigated in the literature [25,26]. In the optical frequencies, the BIC effect is mainly realized on the all−dielectric platform where even the noble metals exhibit high absorption losses. By introducing some symmetry–breaking perturbation into structures supporting the symmetry–protected BIC which happens at the Γ point in the ω–k space (for normal incidence), the ideal BIC will switch to a quasi–BIC (QBIC) mode with finite Q–factor and non–zero linewidth, but with the possibility of being excited by free space radiations [27]. This is currently the most popular design approach to realize high Q factor resonances. Following this strategy, some efforts to explore the use of BIC in the THz band have also been made quite recently. Differently from the scenario in the optical band, most metals exhibit negligible losses in the THz frequencies, so the BIC phenomenon can also be observed in metal−based THz metamaterials [28,29] as well as in all−dielectric structures [30].

In this paper, we present a conceptually different physics to achieve resonances with ultra–high Q factors. A simple structure of composite grating composed of two alternatingly aligned ridge gratings with the same pitch but different ridge widths on a silicon slab, which is lithographically practical and feasible to realize, is used. We note that although the similar symmetry−breaking strategy as in the symmetry−protected BIC is seen, our

working principle is different. The slab waveguide itself supports true guided modes with infinite Q factors over a large operation bandwidth with no access to free space radiations. The gratings only provide a weak modulation of the refractive index so that their presence will only lead to a small bandgap at the first Brillouin Zone (FBZ). However, with the introduction of perturbation (the width of every second ridge is changed), the grating period is doubled, and the FBZ is folded to flip the dispersion line of the original guided modes to be above the light line, forming leaky resonances referred to as quasi–guided modes (QGM) in this work. This is fundamentally different from the physics of BIC, which only happens at very few discrete points in the ω–k space. Thus, even when the BIC is switched to QBIC using intentional perturbation, the operation bandwidth of the QBIC is still quite limited because it is closely concentrated within the spectral band covering the original BIC frequency. In contrast, the QGMs are derived from the guided modes and are intrinsically operational over a large spectral bandwidth.

Similar to the QBIC modes, these leaky resonances of QGM possess ultrahigh Q factors. Compared to a regular ridge grating, this type of composite grating structure supports the QGM resonance at both normal and inclined incidences. The QGM resonance can easily exhibit a Q factor of more than $10^5$ and huge local field enhancement, which is even higher than that provided by optical nanoantennas in the optical frequencies. The electromagnetic characteristics of these QGM resonances in the THz band is comprehensively investigated, laying out the foundation to manipulate and enhance the interactions between THz waves and matters and opening the horizon for novel THz applications in real life.

## 2. Structure and Initial Results

We start by revisiting the regular guided modes supported by a slab waveguide embedded in the background of air and modulated by a ridge–type grating, schematically depicted in Figure 1a. Both the grating and the slab are made from high–resistivity lossless silicon material, whose refractive index is assumed to be 3.418 in the THz band. Only TE mode is considered in this work; i.e., the electric field is along the z direction; however, all the results and conclusions are also valid for the TM mode. The thickness of the Si slab is chosen as t = 88 μm to assume the operation at a low waveguide order. When both the ridge height h and width W are much smaller than the period, e.g., h = 10 μm and W is around 0.1 P, where P is around 100 μm to ensure the resonances around 1.0 THz, the guides modes in the slab waveguide are weakly affected. Figure 1b presents the calculated dispersion lines and the lower two red branches completely below the light lines (the dashed blue lines) correspond to the $TE_0$ guided modes. Very small spectral gaps are present (not quite visible in Figure 1b) at the boundary of the FBZ in the dispersion lines, due to the weak interaction between two counter–propagating guided modes in the slab. This case has been widely investigated in the literature and is the foundation for distributed feedback lasers [31]. At even higher frequencies, the dispersion lines of the guided modes are folded back to be above the light line, as shown by the upper two red lines in Figure 1b, to form the well–known guided mode resonance (GMR), which happens at the second stop band [32]. The occurrence of the GMR is governed by the equation $\frac{2\pi}{\lambda_0}\sin(\theta) + m\frac{2\pi}{P} = \frac{2\pi}{\lambda_0}n_{eff}$ (θ is the incident angle and is 0 for normal incidence, m is an integer, $\lambda_0$ is the working wavelength and $n_{eff}$ is effective refractive index of the slab mode). Apparently, the eigen resonances of the regular GMR are mainly located above the light line, which is also called the continuum region because it represents a collection of continuous spectra of radiating waves. For the resonances within the continuum, the guided mode associated with the GMR will be coupled back to free space by the same grating, observed as the phenomenon of a dip in the transmission spectrum. In the above numerical investigations of the dispersion properties, as well as the subsequent calculation of the grating transmission spectra, a finite element method (FEM)−based commercial software of Comsol Multiphysics is used. When calculating the band diagram, i.e., the eigen frequency versus lateral wave vector $k_x$, the Floquet boundary condition is used to model the lateral periodicity at both normal and inclined incidence.

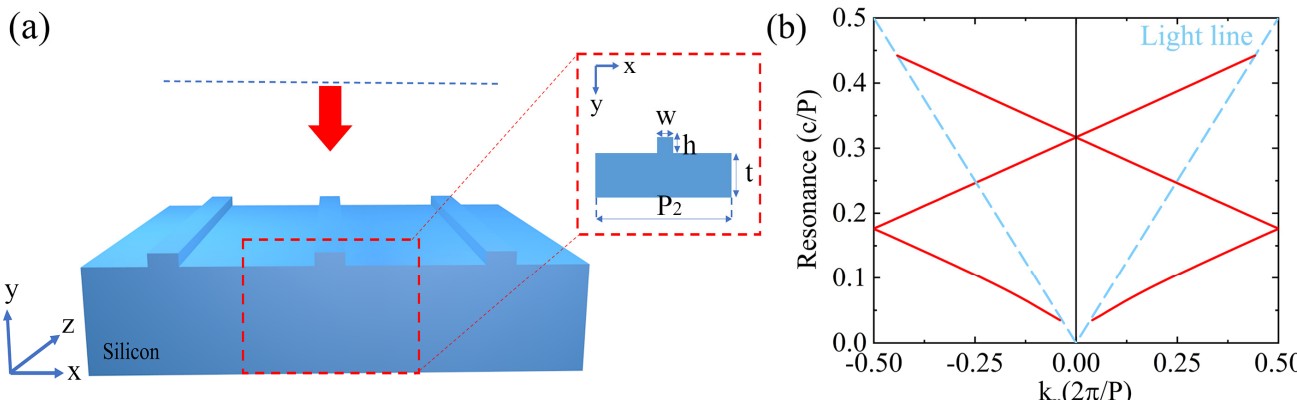

**Figure 1.** (**a**) Schematic of a regular ridge grating on a slab waveguide in air, with inset presenting an enlargement of a single periodic cell. (**b**) Dispersion diagram calculated for $P_2$ = 99 μm and ridge width W of 10 μm. The dashed blue line is for light in free space. The upper two branches above the light line are related to the GMR, which happens around the Γ point corresponding to the second stop band of the grating.

Things will become more interesting when two ridges with slight ridge width difference labeled as δ are used alternatingly, as schematically shown in Figure 2a. We hereafter refer to this special structure as a composite grating. The results of the dispersion curve for the case δ = 1 μm (ridge widths are 10 μm and 11 μm, respectively) are given as the solid red line in Figure 2b. The dashed green line indicates the dispersion of light in the air, and the area above it is the continuum region. The dispersion line for the coupled guided modes supported by the one−ridge grating is also presented as the solid blue line for the purpose of comparison, which is well below the light line and has a small spectral gap at $k_x = \pm\pi/P$. For a non-zero value of δ, the composite grating contains two ridges within the same unit cell, so the period is doubled to $P_2$ ($P_2 = 2P$) and the FBZ shrinks from $(-\pi/P, \pi/P)$ to $(-0.5\pi/P, 0.5\pi/P)$. Since δ is small, the dispersion of the composite grating should have the same profile as the guided modes of the regular grating. Due to the shrinking of the FBZ, the dispersion line of the composite grating can be seen as a folding of that of the guided modes from $k_x = \pi/P$ to $k_x = 0$, i.e., the Γ point. Then, the dispersion lines of the original guided modes with infinite Q factors will now be flipped to be above the light line, suggesting the occurrence of new leaky resonances with the possibility of coupling to free–space radiations, which are referred to as QGMs. Very similar to the well-known QBIC, these QGMs also have ultra−high Q factor which are highly dependent on the level of perturbation. Furthermore, and in huge contrast with QBIC modes, the QGMs retain the spatial dispersion of the original guides modes and thus have a large operation bandwidth. Then, the resonance can be tuned to precisely match a specific frequency by selecting a suitable angle of incidence.

It is known that the regular GMR can also have a large Q factor, especially when the grating has a weak depth and a weak modulation to the refractive index [33]. However, we show that the QGM can have a much higher Q factor than the GMR. To present a straightforward comparison between the GMR and the QGM resonance, we show in Figure 2c the transmission spectrum of a TE (along the z−direction) plane wave normally incident onto the structure. Note that due to the occurrence of the GMR at the second stop band, the period of the structure to support the GMR is doubled to have the resonance close to QGM. The red line is for the case of regular GMR where the ridge width is 10 μm and the period is 97.2 μm. A sharp transmission dip at 973.5 GHz with the full width at half maximum (FWHM) around 0.4 GHz is present. This dip corresponds to the case when the phase matching condition is fulfilled and the grating couples the incident plane wave into the guided mode propagating laterally in the slab. The guided mode will couple back by the same grating structure to free space during its propagation, exhibiting a sharp peak

in the reflection spectrum. For the composite grating composed of two different ridges with the width of 10 µm and 11 µm, a much sharper resonance of the QGM is present on the lower frequency side of the original GMR. The FWHM of this new resonance is found to be around 4 MHz, which is two orders of magnitude smaller than of that for GMR, consistent with the results in the eigen frequency calculations. The red shift in the QGM resonance compared to GMR is due to an increased effective index of the top cladding layer when two ridges instead of one are used.

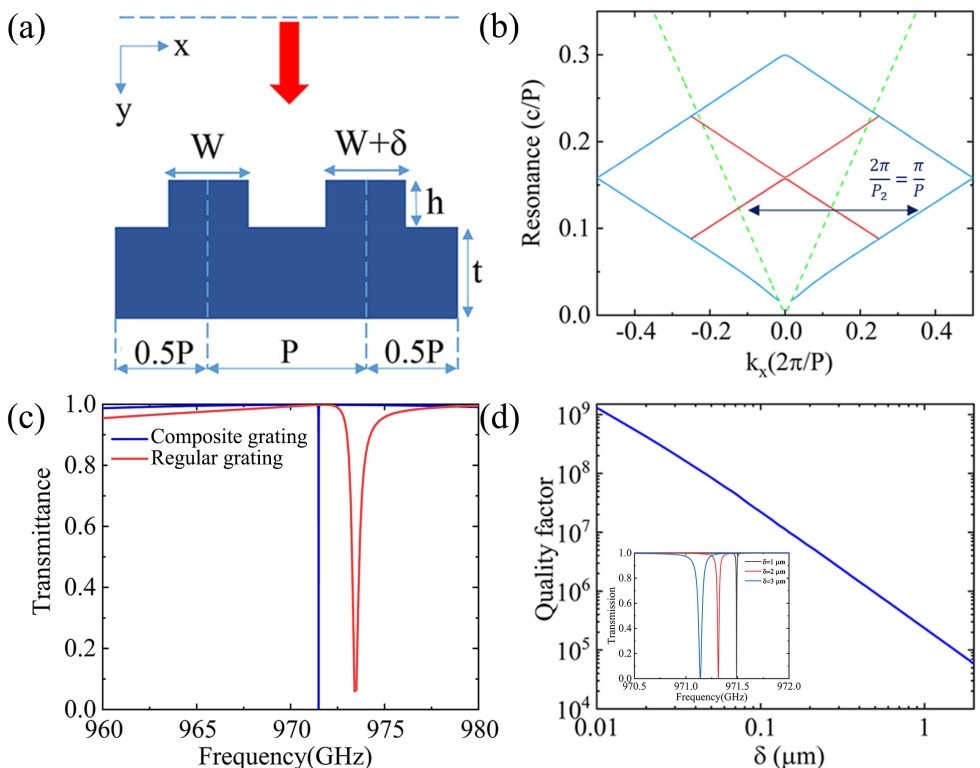

**Figure 2.** (**a**) Schematic diagram of a composite grating with different ridge width on a slab. (**b**) Band structure of the QGM supported by the composite grating. (**c**) Calculated transmission spectrum at normal incidence for a regular GMR (red line) as well as the case for the composite gratings (blue line). (**d**) Quality factor of the QGM as a function of δ, where δ is difference between two gratings. The inset in (**d**) illustrates the transmission spectrum as a function of different values of δ.

Besides the fact that the Q factor of the QGM mode is typically much higher than that of the GMR, what is more attractive is that the Q factor associated with the composite grating exhibits a strong dependence on the value of *δ*. The blue line in Figure 2d presents the Q factor values for $k_x = 0$ as a function of the ridge width difference δ. The quality factor increases substantially for a smaller asymmetry between the ridges, and in general, it is above a level of ∼$5 \times 10^4$, much higher than those achieved for regular GMR. The inset in Figure 2d illustrates the transmission spectrum for the different values of δ, where δ is 1 µm, 2 µm and 3 µm, respectively. The QGM resonances shift from 971.49 GHz to 971.14 GHz can be steadily observed as the ridge width difference δ increases from 1 to 3 µm. As δ increases, the FWHM becomes significantly larger, consistent with the change in Q factor in Figure 2d. In addition, the Q factor approaches infinity when δ shrinks to 0, very similar to the trend of the Q factor of QBIC as a decrease in the perturbation. However, we note that here when δ shrinks to 0, the situation is different from that in BIC. When the width difference δ is 0, the composite grating will switch to one regular grating with the pitch decreasing to P, and then the QGM will switch back to the guided modes whose dispersion is below the light line and can thus not be excited by free space excitations.

## 3. Further Results and Discussions

Having revealed the physical mechanism of the QGM phenomenon supported by the composite grating, we further investigate the general electromagnetic properties of the QGM resonances. For the simplicity of applications, we consider the normal incidence case.

To support the guided mode, the slab structure needs to have a certain thickness to fulfill a lateral (in the Y direction) phase condition. When a plane wave impinges the bare slab structure, due to the existence of the Fabry–Perot (FP) effect, some oscillations will be present in the transmission spectrum, as shown in the inset of Figure 3a. After the composite grating is introduced onto the slab, since the grating ridge is narrow, the FP oscillations are weakly affected and thus working as the background for the QGM resonance. As a result, the QGM resonance may happen at different position of the FP oscillations, and exhibit distinct behaviors. We choose three typical points at the FP resonance, as shown in the inset of Figure 3a: A at the peak where the transmittance through the slab is at its maximum, B at the slope, and C at the bottom of a FP resonance. The spectral position of the QGM resonance at different points can be achieved by choosing a proper value of the grating pitch P. Figure 3a,c,e present the transmission spectrum at normal incidence when the QGM resonance is located at A, B, and C point, by choosing a *p* value of 97.2, 115 and 78 μm, respectively. QGM resonances with extremely narrow bandwidth are observed for all these three cases. To obtain more details of the resonance information, we plot in Figure 3b,d,f the enlarged images of the spectrum close to the respective QGM resonance. The real part of the $E_z$ components are also plotted as the inset for each QGM resonance. Detailed simulation results reveal that for all these three cases, the bandwidth is all at the MHz level. The field distribution of all these QGM resonances are anti–symmetric, which are consistent with the electric field of the guided modes. In cases of A and C, the QGM resonances are located at the maximum/minimum transmittance of the original slab structure, and Lorentzian profiles of the QGM resonances are found. Note that the transmittance at these two QGM resonances drops following the tread line of the original FP background. For example, for case C, the transmittance starts from around 30% to be zero at QGM resonance. In contrast, for case B, where the QGM resonance meets the slope of the FP resonance, there seems to be a strong interaction between the QGM and the FP resonances, and a pronounced Fano profile is seen, with the maximum transmittance reaching 1.0. We further find that the enhancement of the electric field magnitude is the highest for case B, reaching 470, compared to that of 258 and 225 for cases A and C, respectively. The local electric field enhancement is even higher than that provided by spoof surface plasmon–based terahertz antennas [34], suggesting that the QGM effect is very promising in enhancing THz–matter interactions. This high local electric field enhancement is extremely important for terahertz sensing applications, where the absorption of THz radiations is dependent on both the imaginary part of the material permittivity and the local electric field values [35]. The higher sensitivity of the asymmetric Fano profile associated with case B to any refractive index changes within the whole structure also suggests that the Fano type of the QGM resonance is of more interest for practical applications in THz sensing.

We note that all the above results are obtained with periodic boundary conditions in the numerical simulations, i.e., assuming the composite grating extending to infinity in the x direction. So the QGM resonances with ultra–high Q factors are a collective behavior of the periodically aligned elements. In practical applications, both the fabricated device and the incident light beam have limited lateral dimensions, preferably smaller in footprint. To investigate the influence of this finite–size effect, i.e., the spectral behavior of the QGMs at a limited number of ridges, we calculated the transmission spectrum of a Gaussian beam through the composite grating with a limited size. Instead of using a periodic boundary condition, perfectly matched layers are used to truncate the computational domain. In all the simulations, the beam waist radius of the Gaussian beam (over which the electric field decays to the 1/d of its maximum) is fixed to 0.4 $W_x$, where $W_x = NP_2$ is the device size along x direction, and N is the number of grating unit cells, to make sure that the grating accommodates adequately the incident beam. We present in Figure 4 the calculated

transmission spectrum through the finite–size composite gratings with different number of unit cells. It is evident that both the transmission and resonance behavior have a significant dependence on the number of grating elements. When the number is small (e.g., N = 100), the transmittance on resonance does not drop as much as in the periodic structure, due to a lower interaction between the incident beam and the device. Furthermore, the bandwidth is larger. That is because a Gaussian beam can be decomposed into a large number of plane waves with different lateral wavevector components. While each wavevector corresponds to a QGM frequency according to the dispersions in Figure 2b, the transmission spectrum in Figure 4 exhibiting a larger bandwidth at a small number of grating elements and a small Gaussian beam waist is due to the overlap of many QGM resonances. As the number of grating elements increases, the overall behavior approaches that of the infinite structure presented in Figure 2. In general, there must be a large enough number of grating elements to eliminate the finite–size effect and to make sure the collective behavior is significant. For this structure, used to produce the results in Figure 4, the minimum number of elements is in the order of 100. However, this number can be controlled by using a higher modulation of the refractive index by changing the height and width of the ridges.

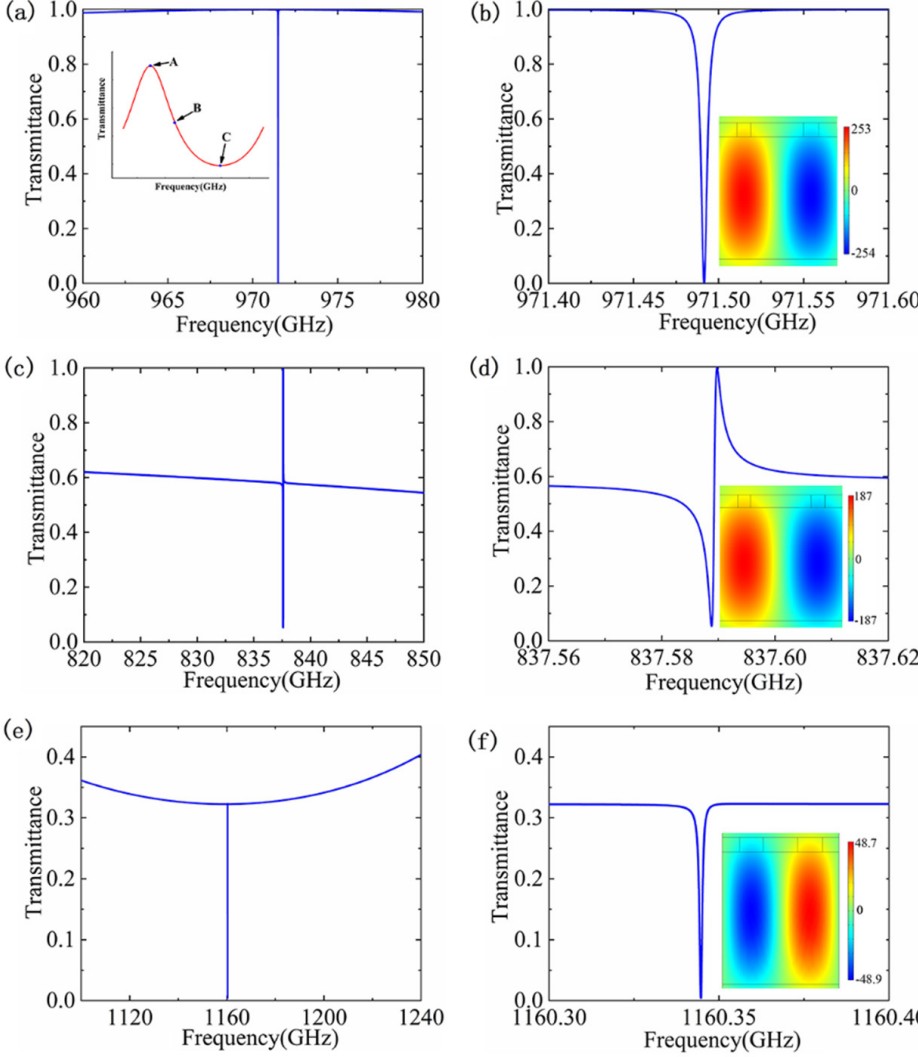

**Figure 3.** Transmission spectra at normal incidence when the QGM resonance happens at the peak (**a**), slope (**c**), and bottom (**e**) of the FP oscillations supported by the bare slab structure. (**b**,**d**,**f**) present the enlarged image close to the QGM resonance, respectively, as well as the real part of $E_z$ distribution across the slab for each QGM. The inset in (**a**) illustrates the transmission spectrum for the FP oscillations, where three points marked as A, B and C labels the position of the FP peak, slope, and bottom.

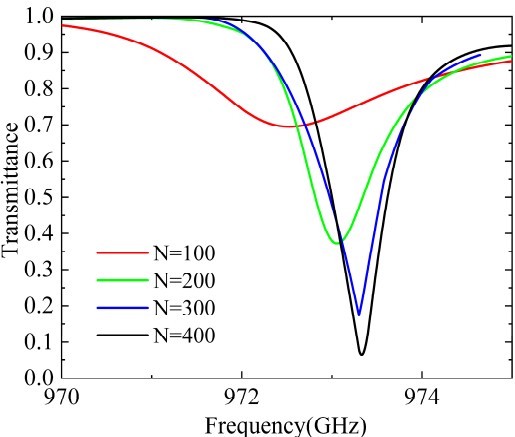

**Figure 4.** Transmission of a Gaussian beam through the composite grating with different number of grating elements. The grating element geometry is the same as before, i.e., W = 10 μm, δ = 1 μm, P = 97.2 μm, t = 88 μm and h = 10 μm.

## 4. Conclusions

One main feature of the QGMs, which also distinguish them from the QBIC modes, is that they inherit the spatial dispersion of the original guided modes. As a result, as one can see from the dispersion lines given in Figure 2b, the QGM dispersions extend over a large operation bandwidth and their resonance frequencies are highly dependent on the lateral wavevector. In other words, the resonance can be tuned by the incident angle of the excitation. This provides an effective means of tuning the resonance frequency without the requirement of using external stimulus, e.g., electric field, and can promise a lot of important applications where the resonance tuning is important.

In summary, in this paper, we have proposed a simple scheme of realizing new leaky modes by exploiting the composite grating structure on a slab. When there is a slight difference in the grating ridge width in the composite grating, it causes the grating period to be doubled and its FBZ to be folded, forming the dispersion of guided mode to be above the light line to QGMs resonance. Ultra–narrow resonance with the Q factors well above $10^4$, and the level of local field enhancement even higher than that provided by spoof surface plasmon–based structures can be steadily achieved for the QGMs at normal incidence. In this work, although the same material as the slab is used for the grating elements, for simplicity and manufacturing considerations, we note that the same concept can be easily extended to other structures, e.g., by using other materials including metals as the grating elements. At low frequencies, since the metal dissipation loss is weak, the radiative loss is the dominant factor affecting the properties of resonating metallic structures. Using metallic grating elements over a dielectric slab provides a new means of achieving high quality resonances using metallic materials. Furthermore, the two dimensional ridges in the composite gratings can also be replaced with more complicated three dimensional elements including a variety of geometries, such as straight rods, elliptical rods, and disks, to name just a few, as long as the asymmetry between two different grating elements can be introduced to change the period and the FBZ in one direction. Illustrative examples of asymmetric metallic structures on a dielectric slab to realize the QGM effect can be found in Figure 5. Special care is required to double the period in one direction when some geometric perturbations are introduced; e.g., the structure in Figure 5 needs to satisfy $P_{2x} = 2P_y$ such that the whole structure becomes a photonic crystal slab (PCS) structure of square lattice when the relative radius $r$ between the two disks is 0. The PCS is known to support GMs, so the QGM effect can be achieved by introducing asymmetric parameters. In addition, the metal material can be gold, silver, etc. In general, a large family of structures can be constructed to achieve the QGM effect. The large local field enhancement provided by this QGM effect renders the investigated structure an outstanding platform to enhance the interactions between matter and terahertz waves, and it helps lay out the foundation

for novel applications in the THz region, even with the limited performances of current THz sources.

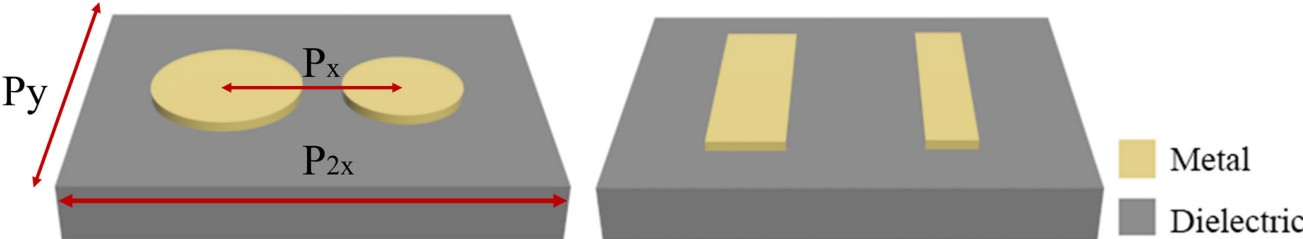

**Figure 5.** Examples of using asymmetric metallic structures such as disks or rods on a dielectric slab to realize the QGMs effect.

As a last remark, we note that although the presented results are achieved as an example in the THz band, the design methodology based on period-doubling perturbation to switch true guided modes (GMs) to QGMs, in order to achieve ultra-high Q factor resonances and enhanced EM wave–matter interactions, is quite generalized; in addition, this strategy can be easily extended to other spectral ranges by using a large variety of period structures, e.g., by using metal stripes on the silicon-on-insulator (SOI) platform in the telecommunication band.

**Author Contributions:** Conceptualization, Z.H.; methodology, Z.H.; software, M.S. and Z.H; validation, Z.H.; formal analysis, M.S. and Z.H.; investigation, M.S.; resources, Z.H.; data curation, M.S.; writing—original draft preparation, M.S. and Z.H; writing—review and editing, Z.H.; visualization, M.S. and Z.H.; supervision, Z.H.; project administration, Z.H.; funding acquisition, Z.H. All authors have read and agreed to the published version of the manuscript.

**Funding:** National Natural Science Foundation of China (11974221, 11974269) and Local science and technology development project of the central government of China (No. YDZX20203700001766).

**Institutional Review Board Statement:** Not applicable.

**Informed Consent Statement:** Not applicable.

**Data Availability Statement:** The data that support the findings of this study are available from the corresponding author upon reasonable request.

**Conflicts of Interest:** The authors declare no conflict of interests.

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
