# Peer review of "Quasi-Guided Modes Supported by a Composite Grating Structure with Alternating Element Widths"

_photonics, doi:10.3390/photonics10020110_

Round 1

Reviewer 1 Report

In this manuscript, the authors have presented a simple structure of composite grating composed of two alternatingly-aligned ridge gratings with the same pitch but different ridge widths on a silicon slab. The proposed structure possesses the ability to enhance wave-matter interactions in the terahertz region by exploiting the phenomenon of quasi-guided modes. They are achieved with high-quality factors and huge local field enhancement. I recommend that this manuscript can be published, but I have some suggestions to improve the quality of the paper.

For my Comments, Please see the attached file. 

Reviewer 2 Report

The Authors propose and investigate a simple configuration based on quasi-guided modes. The results have been carried out by using FEM solver. In my opinion, the configuration could be very interesting mainly to improve the Q-factor, although the quality of the manuscript does not deserve the publication. According to the following comments, I suggest the manuscript rejection.

-          In order to keep the reader attention, the Introduction should be reduced, also highlighting the application of the proposed device. Please use the template for the equations.

-          In the Section 2, the Authors should clarify if the dispersion and losses have been taken into account. Moreover, the overall device is not clear, since Fig. 1 (a) reports just a single period, with features without any design. The design of the features is crucial, also by taking into account the fabrication constraints.

-          Some basic equations, as GMR ones, could be removed.

-          Figure 2 is very confusing. According to (d), the Q-factor decreases as δ increases. Therefore, the maximum Q-factor could be measured for δ  = 0, in contrast to the spectra reported in (c).

-          The Authors give emphasis to Fig. 3. However, it is just the combination of the Fabry Perot effect with resonance spectrum.

-          Figure 4 report a sketch of the device. Are two periods enough to excite a very high-Q factor resonance?

Reviewer 3 Report

This paper reports a study of quasi-guided modes supported by a composite grating structure with alternating element widths. I have some comments.

1. First to sentences of the introduction need improvement in terms of EM for THz and also some words about GHz region. when enumerate some application like environmental or biomedical monitoring is mandatory to add some more references related with the importance not only THz but also GHz region. so, please read and add references very related: IEEE Sensors Journal 21 (3), 3028-3034, 2020; Sensors 22 (11), 4044, 2022; Optical and Quantum Electronics 48 (3), 1-8, 2016.

2. How was optimized the critical values on the simulation? Please add more details.

3. More details about asymmetric metallic structures on a dielectric slab to realize the QGM effect presented in Fig. 4 is needed to be included. Why such structures and not other? Please comment. Also, what kind of metals could be used?

4. Fig 3c and e are not well explained in the text. What this the fundamental effect for such behavior?

Round 2

Reviewer 1 Report

The authors addressed all the suggested comments. 

Author Response

We thank the reviewer for his time in checking our response and his positive recommendation.

Reviewer 2 Report

The Authors have not exhaustively replied to the Reviewer comments. Just few sentences have been added to the manuscript. Moreover, new comments arise: since the dimension of the metasurface is not infinite, how many periods are needed to excite the desired resonance?

I still don't suggest the manuscript publication.

Reviewer 3 Report

Satisfatory revision

Author Response

(The authors gave the same response as above.)
